# Can Pruning Improve Certified Robustness of Neural Networks?

**Zhangheng Li, Tianlong Chen**                               *zoharli, tianlong.chen@utexas.edu*
*Department of Electrical and Computer Engineering*
*University of Texas at Austin*

**Linyi Li, Bo Li**                                          *linyi2, lbo@illinois.edu*
*Department of Computer Science*
*University of Illinois Urbana-Champaign*

**Zhangyang Wang**                                           *atlaswang@utexas.edu*
*Department of Electrical and Computer Engineering*
*University of Texas at Austin*

**Reviewed on OpenReview:** *https://openreview.net/forum?id=6IFi2soduD*

## Abstract

With the rapid development of deep learning, the sizes of deep neural networks are getting larger beyond the affordability of hardware platforms. Given the fact that neural networks are often over-parameterized, one effective way to reduce such computational overhead is neural network pruning, by removing redundant parameters from trained neural networks. It has been recently observed that pruning can not only reduce computational overhead but also can improve empirical robustness of deep neural networks (NNs), potentially owing to removing spurious correlations while preserving the predictive accuracies. This paper for the first time demonstrates that pruning can generally improve $L_\infty$ certified robustness for ReLU-based NNs under the *complete verification* setting. Using the popular Branch-and-Bound (BaB) framework, we find that pruning can enhance the estimated bound tightness of certified robustness verification, by alleviating linear relaxation and sub-domain split problems. We empirically verify our findings with off-the-shelf pruning methods and further present a new stability-based pruning method tailored for reducing neuron instability, that outperforms existing pruning methods in enhancing certified robustness. Our experiments show that by appropriately pruning an NN, its certified accuracy can be boosted up to **8.2%** under standard training, and up to **24.5%** under adversarial training on the CIFAR10 dataset. We additionally observe the possible existence of *certified lottery tickets* in our experiments that can match both standard and certified robust accuracies of the original dense models across different datasets. Our findings offer a new angle to study the intriguing interaction between sparsity and robustness, i.e. interpreting the interaction of sparsity and certified robustness via neuron stability. Codes will be fully released.

## 1 Introduction

Neural Network (NN)-based framework is a strong general solution to many problems, yet many of these solutions remain impractical for real-world applications of low fault tolerance. A slight perturbation in the raw input sensory data could completely change the predicting behaviors of the networks. Furthermore, researchers have shown that various kinds of targeted adversarial attacks can easily fool the neural networks Szegedy et al. (2013); Goodfellow et al. (2014), which poses threat to many deep learning applications. Fortunately, researchers introduced formal methods to verify neural network behaviors, which made it possible to mathematically derive the prediction bound of a neural network w.r.t. a certain input, and thus evaluate

the certified robustness of the neural network. For example, in the image classification task, given an input image with some perturbation, if the lower bound of the output probability of the correct label is higher than the upper bound of probabilities of other incorrect labels, we say the model is certifiably robust w.r.t. this image sample. The goal of the neural network verification is to estimate the bounds as accurate as possible.

In this paper, we are concerned about the certified robustness w.r.t. $L_\infty$ input perturbations under the *complete verification* setting, where the verifier should output the *exact* bounds given the input domain $\mathcal{C}$, rather than some relaxation of $\mathcal{C}$, given sufficient time. Despite its theoretical appeal, the complete verification of neural networks is known to be a challenging NP-hard problem Katz et al. (2017); Weng et al. (2018), mainly due to the non-linear activation functions in neural networks, such as Sigmoid and ReLU. The popular *Branch-and-Bound* (**BaB**) framework Bunel et al. (2017) utilized the feature of ReLU activations and adopted the classical divide-and-conquer method to solve the complete verification problem. It branches the bound computation into multiple sub-domains recursively on ReLU nodes and computes the bounds on each sub-domain respectively. The time complexity of this framework is exponential, and typically has pre-set time limit for each sample verification.

Several verifiers Xu et al. (2020b); Wang et al. (2021) based on the BaB framework were later proposed for efficient complete verification. The core problem addressed in these methods is how to estimate the pre-activation bound (i.e. propagated input bounds of the non-linear activation layers) as tight as possible given limited time. To approach this, they use Linear-Relaxation based Perturbation Analysis (LiRPA) to relax non-linear bound propagation with linear ones and use GPU-accelerated BaB methods to further tighten the estimated bounds as much as possible. However, the estimated bound is still loose, mainly because (1) an efficient linear relaxation of multiple non-linear activation layers is loose both empirically Salman et al. (2019) and theoretically Katz et al. (2017); Weng et al. (2018), where tightening the relaxation requires an exponential number of linear constraints which is inefficient Tjandraatmadja et al. (2020); and (2) the BaB framework requires solving an exceedingly large number of sub-domains (which is exponential in the worst case Katz et al. (2017)) to provide a tight bound, so in practice we often solve only a part of the sub-domains which yields a loose bound.

Recent efforts Fu et al. (2021); Gui et al. (2019); Hu et al. (2020); Ye et al. (2019); Jordao & Pedrini (2021); Xiao et al. (2018) reveal that proper network pruning can empirically enhance neural network robustness to adversarial attacks. We take one step further to argue that pruning can also be utilized to improve the estimated bound tightness of certified robustness verification, by alleviating linear relaxation and sub-domain split problems. Improving empirical robustness (as a surrogate of "ground-truth" robustness) and verification tightness can together lead to overall measurable certified robustness, and we find that existing pruning schemes can already co-achieve both. Moreover, inspired by the fact that sparsity can eliminate unstable neurons and improve non-linear neuron stability for verification Xiao et al. (2018), we present a new stability-based pruning method, that even outperforms existing pruning methods on improving certified robustness. Our main contributions are outlined below:

- For the first time, we demonstrate that pruning can generally improve *certified* robustness w.r.t. $L_\infty$ input perturbations. We analyze pruning effects from the perspectives of both improving ground-truth robustness of the model and the verification bound tightness, and empirically validate it with extensive pruning methods and training schemes.

- As pruning can be utilized to improve non-linear neuron stability, we propose a novel regularizer for pruning called *NRSLoss* (see Figure 2) that effectively regularizes the neuron stability and outperforms existing pruning methods in enhancing certified robustness.

- Our experiments validate the above proposals by presenting verification results under various settings. For example on the CIFAR10 dataset, under certified training, existing pruning methods as well as our proposed NRSLoss-based pruning boost the certified accuracy for 1.6-7.1% and 8.2% respectively; under adversarial training, they boost the certified accuracy for 12.5-24.5%.

## 2 Related Work

### 2.1 Incomplete and complete NN verification

Neural Network verification is a critical issue for developing trustworthy and safe AI. Existing verifiers can be divided into either *incomplete* or *complete* verifiers. Complete verifiers can typically produce tighter bound than incomplete verifiers, but consume much larger computational resources than incomplete verifiers. Typical incomplete verifiers are based on duality Dvijotham et al. (2018); Raghunathan et al. (2018) and linear approximations Weng et al. (2018); Wong & Kolter (2018); Zhang et al. (2018a), whereas existing complete verifiers are based on satisfiability modulo theories (SMT) Katz et al. (2017); Ehlers (2017), mixed integer programming (MIP) Tjeng et al. (2017), convex hull approximation Müller et al. (2021), or Branch-and-Bound (BaB) Bunel et al. (2017); Xu et al. (2020b); Wang et al. (2021).

Traditional complete verifiers such as SMT and MIP are computationally expensive and hard to parallelize. To this end, a series of verifiers based on the BaB framework were recently proposed for efficient and parallelizable complete verification. Auto-LiPRA Xu et al. (2020a) was an early incomplete verifier and certified trainer, that relaxes ReLU non-linearity with a tight linear relaxation. Following that, Fast-and-Complete algorithm Xu et al. (2020b) proposed to combine auto-LiRPA with BaB for tighter bound estimation and use LP solver for completeness check, which is a GPU-parallelizable complete verification method. Beta-CROWN Wang et al. (2021) further extended Fast-and-Complete algorithm by replacing the LP completeness check with optimizable constraints based on the Lagrange function, and improved the verification efficiency.

### 2.2 Neural network pruning

Pruning removes the redundant structures in NNs and reduces the size of parameter numbers from the computation graph of NNs. It not only constitutes an important class of NN model compression methods but also can act as a regularizer for NN training which can improve the performance w.r.t. original unpruned networks. The pruning process can be conducted at different levels or granularities, such as parameter-level Frankle & Carbin (2018); Zhang et al. (2018b); Ma et al. (2020), filter-level Luo et al. (2018); Li et al. (2016); Roy et al. (2020) and layer-wise Wang et al. (2018); Wu et al. (2018); Zhang et al. (2019). Especially, Lottery Ticket Hypothesis (LTH) Frankle & Carbin (2018) claims the existence of independently trainable sparse subnetworks that can match or even surpass the performance of dense networks. Such sparse subnetworks called "winning tickets", can be obtained by simple iterative parameter-level pruning.

### 2.3 Pruning and robustness

Recently, several works Ye et al. (2019); Gui et al. (2019); Jordao & Pedrini (2021) have revealed that proper network pruning can empirically improve the robustness of a trained NN, potentially due to removing spurious correlations while preserving the predictive accuracies. Fu et al. (2021) found that randomly initialized robust subnetworks with better adversarial accuracy than dense model counterparts can be found by IMP. Xiao et al. (2018) was the first work to inject sparsity during NN training, with the primary goal to *speed up* certified verification. It considered only weight magnitude pruning, and did not generally demonstrate sparsity to *improve* the achievable $L_\infty$ certified robustness. HYDRA Sehwag et al. (2020) incorporated the robustness loss as a pruning objective, and showed such a robustness-aware pruning scheme can lead to high NN sparsity without much robust accuracy loss. Besides studying incomplete certified verification, HYDRA did not specifically analyze what benefits pruning brings for verification, while our NRSLoss-based pruning is explicitly motivated by reducing unstable neurons and tightening the estimated bound in the complete verification. Han et al. (2021) proposed that *superficial* neurons that contribute significantly to the feature map values in shallow layers were highly localized and are more prone to *adversarial patches*. Hence they used pruning to remove superficial neurons and improved certified defense against adversarial patches – an orthogonal goal to our work.

# 3 Methodology

In this paper, following prior works on certified robustness, we focus on non-linear feed-forward neural networks with ReLU activations. In this section, we first analyze what benefits would network pruning brings to certified robustness and verification, and then introduce the specific pruning methods we test in our experiments.

## 3.1 Preliminary

### 3.1.1 Unstructured and Neuron Pruning

Network pruning is one of the most effective model compression paradigms for deep neural network, by removing redundant parameters or neurons from over-parameterized neural networks, and many of them focus on pruning learnable weight parameters. Existing weight pruning methods can be divided to unstructured pruning and neuron pruning, depending on whether weights are pruned individually or by group. For unstructured pruning, individual weights that connect two channels (neurons) of adjacent layers are removed; for neuron pruning, all input and output weights associated with certain channels are removed.

Each of these two pruning paradigms has its advantages and disadvantages over the other. Unstructured pruning can better preserve the performance of the original dense networks due to pruning flexibility on individual weights, but is hard to realize real hardware acceleration during inference. In contrast, the hardware compression during inference for neuron pruning can be easily implemented due to the removal of entire channels, but has less pruning flexibility compared to unstructured pruning, which would generally lead to worse performance that unstructured pruning. In this paper, we investigate the influence of both unstructured and neuron pruning on certified robustness and are interested in both the performance gain and the reduction of computational overhead brought by pruning.

### 3.1.2 Lottery Ticket Hypothesis(LTH)

The lottery ticket hypothesis Frankle & Carbin (2018) states that a randomly initialized dense neural network contains at least one subnetwork (i.e. by pruning the parameters of the dense network) that has the same initialization of the unpruned parameters and can match the test performance of the dense network after training for at most the same iterations as the dense network, and such subnetworks are called the winning tickets of the dense network. To find these winning tickets, they propose **Iterative Magnitude Pruning**(IMP) algorithm: Firstly, start from a dense initialization $W_0$, and then train the network until convergence to weight $W_t$. Then we determine the $\rho$ percent smallest magnitude weights in $|W_t|$ and create a binary mask $m_0$ that prunes these. Then retrain the pruned network from the same initialization weight $W_0 \odot m_0$ to convergence. Iterating this procedure will produce subnetworks with different sparsity, among which certain subnetworks can match the test performance of the original dense network, i.e. the winning tickets.

### 3.1.3 ReLU Neuron Stability

The illustration of ReLU neuron stability is demonstrated in Figure 1. The ReLU activation function is zero when input value is less than 0, and an identity function when input value is no less than 0. As shown in the Figure 1, $\mathbf{h}_j^{(i)}$ means the pre-activation value of $j$th ReLU neuron at $i$th layer of the network. $\mathbf{g}_j^{(i)}$ means the corresponding value after passing the ReLU neuron. $\mathbf{l}_j^{(i)}$ and $\mathbf{u}_j^{(i)}$ refers to the lower bound and upper bound of the pre-activation $\mathbf{u}_j^{(i)}$ w.r.t. certain input perturbation. Figure 1(a) and Figure 1(d) are unstable neurons where $\mathbf{l}_j^{(i)}$ and $\mathbf{u}_j^{(i)}$ has different signs, while Figure 1(b) and Figure 1(c) are stable neurons where $\mathbf{l}_j^{(i)}$ and $\mathbf{u}_j^{(i)}$ has the same sign. The yellow areas in Figure 1(a) and Figure 1(d) refer to the bounded area of "triangle" relaxation and linear relaxation, respectively. As shown in Figure 1(a), the lower and upper bounds of $\mathbf{g}_j^{(i)}$ will be the intersection of the orange line and the bound of the yellow triangle, and the horizontal location of the orange line will vary from $\mathbf{l}_j^{(i)}$ to $\mathbf{u}_j^{(i)}$ as the input varies within the given perturbation scale. The drawback of the triangle relaxation during bound propagation is that the lower bound of $\mathbf{g}_j^{(i)}$ is not linear.

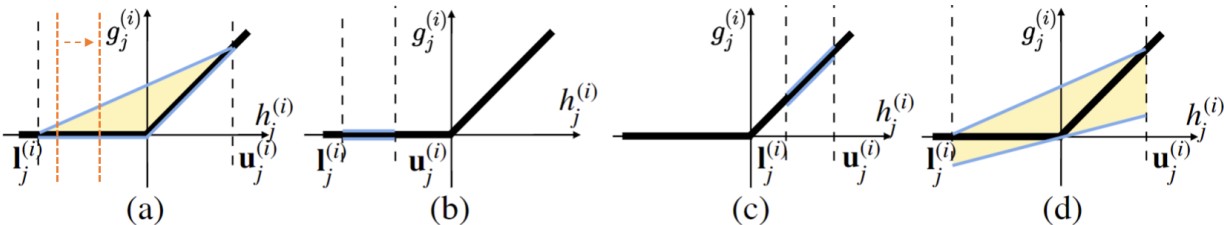

Figure 1: (Figure source from Xu et al. (2020b))) Illustration of the stability of ReLU neuron and linear relaxation. For certain network input with perturbations, (b) and (c) demonstrate stable neurons since their lower and upper pre-activation bounds are on the same side of the Y axis, while (a) and (d) demonstrate unstable neurons since the two bounds are on the different sides of the Y axis. (a) and (d) also show different relaxation methods: (a) is "triangle" relaxation and (d) is the LiRPA-based linear relaxation Xu et al. (2020a).

LiRPA-based linear relaxation (1(d)) addresses this issue by replacing the non-linear lower bound with an dynamic linear lower bound which is a line passing through the origin with an optimizable slope between [0,1]. For more details about neuron stability and bound relaxation, please refer to Xu et al. (2020b).

## 3.2 What factors influence $L_\infty$ certified robustness?

Up to now, with state-of-the-art robust training method and certified verifier, the measurable bound of a trained non-linear neural network is influenced by two major factors:

### 3.2.1 The ground-truth bound of the network

This is mainly decided by the training method. For example, certified training usually provides much higher certified robustness than adversarial training, which is demonstrated in our experiments. In this paper, we ideally hope pruning has positive or no influences on the normal training process, e.g. the influences of pruning-related regularizer to normal gradient back-propagation. Besides, the model size, i.e. the parameter number also matters, which is important in this paper since we can use pruning to reduce parameter number. Intuitively, with more parameter numbers, the bound of the network output tends to be looser. With network pruning, we can heuristically tighten the bound of the network since the network has fewer parameters.

### 3.2.2 Estimated bound tightness of the verifier

This refers to the closeness of the estimated bound of the verification to the ground truth bound of the network, and it reflects the performance of the verifier. Since existing certified verifiers usually have a very large computational overhead to verify even a single sample, in practice, we are concerned about the bound tightness a verifier can reach given a limited verification time.

Although efficient certified verification for non-linear neural networks has made rapid progress in recent years, the bound tightness, running speed and affordable network capacity on limited hardware resources still have huge space awaiting to be improved. Specifically, we identify two major problems that influence the bound tightness:

- **Neuron Stability and Verification Speed.** As mentioned in the introduction, BaB is the mainstream framework of existing certified verification methods. The bound tightness is largely influenced by how many unstable neurons have been branched given a limited time. (Please refer to Section 3.1.3 for the explanation of neuron stability). The verifier would need to visit more sub-domains by branching on unstable neurons within limited time, and thus the verification speed matters.

- **Linear Relaxation.** To facilitate bound computation, many recent verifiers Wang et al. (2021); Xu et al. (2020b); Wong & Kolter (2018); Zhang et al. (2018a); Singh et al. (2019), utilize the linear relaxation method for unstable ReLU neurons, as we illustrated in Section 3.1.3. Normally, if an

unstable neuron has not been branched by BaB, then this neuron gets linearly relaxed during bound propagation, which will also loosen the bound tightness of the verifier.

### 3.3 What benefits for certified robustness can we expect from pruning?

Network pruning can bring many benefits to problems as mentioned in Section 3.2, outlined as follows:

- **Reducing parameter number**. It can tighten bound propagation directly. Both neuron and unstructured pruning can reduce parameter number and the bound propagation becomes tighter after pruning. Take the widely used bound propagation method for linear layers(e.g. convolution and full-connected layer)——Interval Bound Propagation (IBP) for computing $L_\infty$ bounds as an example, its computation can be formulated as follows:

$$
\begin{aligned}
\mu_{i-1} &= \frac{\bar{z}_{i-1} + \underline{z}_{i-1}}{2} \\
r_{i-1} &= \frac{\bar{z}_{i-1} - \underline{z}_{i-1}}{2} \\
\mu_i &= \mathbf{W}\mu_{i-1} + b \\
r_i &= |\mathbf{W}|r_{i-1} \\
\underline{z}_i &= \mu_i - r_i \\
\bar{z}_i &= \mu_i + r_i
\end{aligned}
\tag{1}
$$

  Eq. 1 computes the linear bound propagation for $i$th linear layer of the network, where $\bar{z}_{i-1}$ and $\underline{z}_{i-1}$ are the input lower and upper bounds of $i$th linear layer, and $\bar{z}_i$ and $\underline{z}_i$ are the corresponding output lower and upper bounds, respectively. $\mathbf{W}$ and $b$ denote the weight and bias of the linear layer. The difference between output upper and lower bound equals $2|\mathbf{W}|r_{k-1}$. By network pruning, the weight matrix $\mathbf{W}$ becomes sparser, and thus the difference between output upper and lower bound tends to become smaller, and thus the overall output bound of the network would be tightened.

- **Reducing unstable neurons**. By reducing the number of unstable neurons, we can reduce the number of linear relaxations and sub-domain splits needed by the verification process, which can also directly tighten the bound and accelerate the verification process as well. However, for most existing pruning methods, reducing unstable neurons is not an explicitly designated goal, but rather a possible side effect.

- **Real hardware acceleration with structural sparsity**. If adopting neuron pruning, we can eliminate channels which will concretely reduce the network width on the hardware implementation level. This can accelerate verification and even make resource-intensive verification possible on larger models.

Those possible benefits are further entangled with each other. For empirical evaluation of these benefits, we simply follow the classical criteria: to evaluate the certified accuracy, time and memory consumption, and network width/depth that can be verified after pruning.

### 3.4 Pruning methods

In this section, we test a range of off-the-shelf pruning methods for improving certified robustness. For each pruning method, unless otherwise mentioned, we combine it with iterative pruning with weight rewinding Frankle & Carbin (2018), as we find iterative pruning with weight rewinding generally enhances performance compared to finetune-based pruning or one-shot pruning in our experiments.

#### 3.4.1 Existing pruning methods

In unstructured pruning, for simplicity, we only prune the weights of convolutional layers and ignore linear layers. We pick several representative methods including: 1) **Random Pruning**: pruning weights randomly,

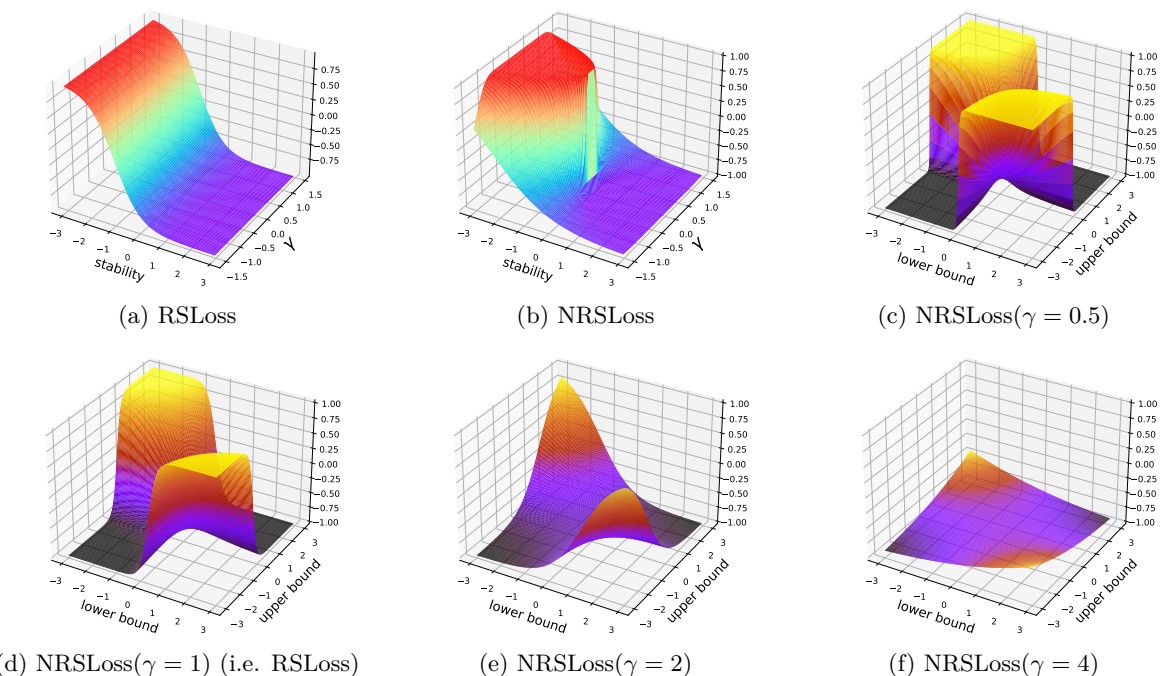

(a) RSLoss     (b) NRSLoss     (c) NRSLoss($\gamma = 0.5$)

(d) NRSLoss($\gamma = 1$) (i.e. RSLoss)     (e) NRSLoss($\gamma = 2$)     (f) NRSLoss($\gamma = 4$)

Figure 2: (a)(b) The landscape of RSLoss and NRSLoss with varied stability and BN channel weight $\gamma$. *Stability* means the stability of a ReLU neuron, i.e. the pre-activation lower bound times upper bound. $\gamma$ is the corresponding channel weight of Batch-Normalization layer, whose magnitude denotes the importance of each channel. The NRSLoss is high when the neuron is unstable and the corresponding channel has low importance. (c)-(f) The sample landscape of NRSLoss with varied lower and upper pre-activation bounds given different fixed $\gamma$. With the growth of $\gamma$ that implies channel(neuron) importance, the NRSLoss gets increasingly suppressed.

which is used for sanity checks in our experiments. 2) Lottery ticket hypothesis (LTH), or denoted as **IMP** Frankle & Carbin (2018): pruning weights with the smallest magnitudes, the most standard pruning scheme. 3) **SNIP** Lee et al. (2018): pruning weights with least loss sensitivity w.r.t. percentile magnitude change. 4) **TaylorPruning (TP)** Evci (2018): saliency-based pruning via a first-order Taylor approximation. 5) **HYDRA** Sehwag et al. (2020): learnable mask-based pruning that minimizes the robustness loss empirically. The original HYDRA pruning uses adversarial loss to select the pruning mask, in certified training setting, we will adapt the adversarial loss to certified loss.

For neuron pruning, we choose two methods: 1) **StructLTH** Anonymous (2022): a structured variant of LTH recently proposed, by using IMP first then ranking channels by their total magnitudes in remaining weights from high-to-low. Then we prune lowest-ranked channels and refill the IMP-pruned weights in the remaining channels. 2) **Network Slimming** Liu et al. (2017): for batch normalization (BN) Ioffe & Szegedy (2015) layers, we have:

$$y = \frac{x - E[x]}{\sqrt{Var[x] + \epsilon}} * \gamma + \beta \tag{2}$$

Network slimming enforces the L1-norm regularizer on $\gamma$ and prune channels with the smallest $\gamma$ magnitudes.

### 3.4.2 Stability-based Pruning

In the context of certified robustness, we wish to eliminate unstable neurons as much as possible, such that the estimated bound tightness of the verification can be improved. Pruning is a natural choice to accomplish this goal. Next, we first introduce a criterion that measures the degree of neuron stability of a given network, and

then introduce an effective stability-based regularizer and corresponding unstructured and neuron pruning methods.

Formally, we denote $[\mathbf{l}_j^{(i)}, \mathbf{u}_j^{(i)}]$ as *bound interval* of the pre-activation value of $j$-th ReLU neuron at $i$-th layer. We want to measure not only the number of unstable ReLU neurons with bound intervals crossing the zero point but also to what extent the instability is. To address this problem, we propose to use $-\mathbf{l}_j^{(i)} \cdot \mathbf{u}_j^{(i)}$ to measure the degree of instability of this neuron. It is easy to know that if we keep the bound interval width $\mathbf{u}_j^{(i)} - \mathbf{l}_j^{(i)}$ unchanged, then $-\mathbf{l}_j^{(i)} \cdot \mathbf{u}_j^{(i)}$ reaches maximum when $\mathbf{l}_j^{(i)} = -\mathbf{u}_j^{(i)}$, which means maximal instability given the same bound interval width. And thus, we simply use the average of this criterion to denote the total degree of instability of the network:

$$instability = \sum_i \sum_j -\mathbf{l}_j^{(i)} \cdot \mathbf{u}_j^{(i)} \tag{3}$$

Similar to this criterion, Xiao et al. (2018) proposed a regularizer named *RS Loss* to regularize ReLU stability and improved certified robustness. The RS Loss is defined as:

$$l_j^{rs} = -\tanh(1 + \mathbf{l}_j \cdot \mathbf{u}_j) \tag{4}$$

This loss can be naturally used to optimize the instability criterion, as the *tanh* wrapper provides smooth gradients. However, we empirically find that for deep networks, batch-normalization (BN) Ioffe & Szegedy (2015) layers, which are placed before ReLU layers, are necessary for the training convergence of pruned subnetworks. In this way, the performance of the RS Loss regularizer is insignificant, because it passes gradients to $\gamma_j$ (i.e. the BN weight of channel $j$) and affects the training process, and the optimization space is relatively small due to the BN constraint. However, we find the pre-BN bounds (i.e. the input bounds of BN layers) to be very flexible. Thus, instead of regularizing the pre-activation bounds using RS Loss, we propose an alternative of *Normalized RS Loss* (**NRSLoss**) to directly regularize the pre-BN bounds, normalized from pre-activation bounds:

$$l_j^{nrs} = -\tanh(1 + \frac{\mathbf{l}_j \cdot \mathbf{u}_j}{\gamma_j^2}). \tag{5}$$

Since the magnitudes of $\mathbf{l}_j$ and $\mathbf{u}_j$ are scaled from pre-BN bounds by the factor of $\gamma_j$, NRSLoss essentially computes RS Loss on the pre-BN bounds $\mathbf{l}_j/\gamma_j$ and $\mathbf{u}_j/\gamma_j$. During training, the NRSLoss is combined with the original loss with an empirical coefficient. Note that we **stop** the gradients back-propagated from NRSLoss to $\gamma_j$, to ensure stable training, especially for pruned subnetworks. Since NRSLoss uses IBP to compute the neuron bounds, it has the same computational complexity as a forward pass of the neuron network. Therefore, it does not bring extra complexity to the training and pruning process. It also has no influence for verification since NRSLoss regularizer will only be used during iterative training and pruning stage.

The loss landscape of NRSLoss is shown in Figure 2. Note that the *stability* term is essentially $0 - instability$. From the loss landscape, we can interpret NRSLoss from another perspective: it penalizes neurons with high instability and low channel importance; when the instability increases, it takes larger channel importance to suppress NRSLoss.

We empirically find that combining NRSLoss as a training regularizer with pruning weights based on the least weight magnitude criterion is the most effective in most cases. We call this pruning scheme IMP+NRS: we first apply NRSLoss during training to regularize the network stability; then we apply magnitude-based pruning to remove weights that are not important for both robustness loss and stability loss.

We stress the two-fold novelty of NRSLoss as follows:

- It takes into account both neuron importance and stability (see Figure 2). In NRSLoss, the magnitude of BN channel weight $\gamma_j$ denotes the importance of each channel. The NRSLoss is high when the neuron is unstable and the corresponding channel has low importance. In contrast, the RSLoss is

irrelevant to channel importance, which might lead to imposing too much regularization on important neurons.

- It also disentangles the influence of stability regularization with the BN layers, via eliminating the magnitude scaling effect to the pre-activation bounds brought by the channel weight $\gamma$. In this way, BN layers can still be learned normally to control the gradients and thus it causes significantly less negative influences to the training performance.

In our experiments, we show in Figure 5 that both RSLoss and NRSLoss can reduce network instability, however, since RSLoss only optimizes towards network stability while NRSLoss takes into account of both neuron importance and stability to avoid imposing too much regularization on important neurons, and causes less negative influences to the gradient flow of BN layers. Therefore, NRSLoss-based pruning can always achieve better performance than RSLoss-based pruning as we show in experiments.

## 4    Experiments

In this section, we evaluate all introduced pruning methods with different training schemes and perturbation scales, and try to address three major questions: *(1) Can existing pruning methods improve $L_\infty$ certified robustness generally? (2) How can NRSLoss-based pruning improve certified robustness? (3) Can we find the existence of certified lottery tickets, i.e., sparse subnetworks after pruning that can restore not only the original performance but also certified robustness?*

Based on our experiment findings, we also provide more ablation studies to further rationalize our claims. In light of whether pruning keeps effective for certified robustness at scale, we conduct experiments on the downscaled 64x64 ImagenetChrabaszcz et al. (2017) with incomplete verifications, please refer to Appendix D for more details. Finally, we briefly summarize our experimental findings with several interesting takeaways.

### 4.1    Experiment Setup

Table 1: The feedforward model architecture in our experiments. ConvBlock(*in*,*out*,*k*,*s*) refers to the composition of (convolution layer, BN layer, ReLU layer) where the convolution layer has *in* input channels, *out* output channels, $k \times k$ kernel size and *s* strides. Note that for the FashionMNIST task which takes greyscale images instead of RGM images as input, we modify the input channel number of the first convolutional layer from 3 to 1, and modify the input units of the first FC layer from 2048 to 1152 accordingly. The parameter count of this model is 6.7M.

| INPUT |
|:---:|
| CONVBLOCK(3,32,3,1) |
| CONVBLOCK(32,64,4,2) |
| CONVBLOCK(64,64,3,1) |
| CONVBLOCK(64,128,4,2) |
| CONVBLOCK(128,128,4,2) |
| FC(2048,100) |
| RELU |
| FC(100,10) |
| OUTPUT |

#### 4.1.1    Dataset and Network architecture

Across our experiments, we choose 3 classification datasets, FashionMNIST Xiao et al. (2017), SVHN Netzer et al. (2011), and CIFAR10 Krizhevsky et al. (2009) as the benchmark datasets. We introduce them as follows:

- **FashionMNIST**: FashionMNIST is an MNIST-like greyscale image classification dataset by replacing hand-written digits with fashion items, which are more difficult to classify. It has a training set of 60,000 examples and a test set of 10,000 examples. Each example is a 28x28 grayscale image, associated with a label from 10 classes. We use the first 200 samples from the testing dataset for verification.

- **SVHN**: SVHN is a dataset consisting of Street View House Number images, with each image consisting of a single cropped digit labeled from 0 to 9. Each example is a 32x32 RGB image. We use the first 200 samples from the testing dataset for verification.

- **CIFAR10**: CIFAR10 is a dataset consisting of 10 object classes in the wild, and each class has 6000 samples. This dataset is commonly used in prior works in complete verification, following Wang et al. (2021), we choose the ERAN test set Singh et al. (2019) which consists of 1000 images from the CIFAR10 test set. Note that we only use the first 200 samples in the ERAN test set for verification efficiency.

Note that we don't use larger datasets such as CelebA Liu et al. (2015) and Imagenet, mainly because training on larger datasets typically require larger network size to achieve good performance. For our choosed datasets, we use a 7-layer CNN as the benchmark model, whose architecture is shown in Table 1. It follows the standard feed-forward CNN structure as in prior works on complete verification Xu et al. (2020b); Wang et al. (2021); Shi et al. (2021). This model has 6.7M parameters and is the largest network that can be fitted in a GPU with 24GB memory for complete verification and can also achieve good baseline performance on our chosen benchmarks. We empirically find that varying the network depth and width doesn't significantly change the relative performance of different pruning methods, therefore we only use this largest network across our experiments for considerations of both model scale and hardware resource constraint.

The design follows the *cifar10-model-deep* setting in Wang et al. (2021), but is wider, deeper, and has BN layers.

### 4.1.2 Pruning methods

We try different setups of hyperparameters for unstructured and neuron pruning. For unstructured pruning methods, we follow the default setting in Frankle & Carbin (2018) and set the iterative weight pruning rate to 0.2, and prune 16 times with re-training; for neuron pruning methods, we keep a similar pruning speed. [1]

We set the NRSLoss weight to 0.01 and set the L1-norm regularizer weight to 0.0001 following Liu et al. (2017). For RS Loss and NRS Loss-based unstructured pruning, we train with these losses and pruning with IMP, as Section 3 introduced, denoted as *IMP+RS* and *IMP+NRS*. We also test training with NRSLoss and pruning with HYDRA, denoted as HYDRA+NRSLoss, to demonstrate the effectiveness of NRSLoss as a regularizer.

### 4.1.3 Training methods

To demonstrate the general effectiveness of network pruning, we choose SOTA adversarial and certified training methods:

**Adversarial training**: we choose the advanced FGSM+GradAlign Andriushchenko & Flammarion (2020) as the adversarial training method (we denote it as FGSM for conciseness hereinafter). The learning rate is set to 0.01, and we use Stochastic Gradient Descent with 0.9 momentum and 0.0005 weight decay as optimizer. All GradAlign-related hyperparameters follow Andriushchenko & Flammarion (2020).

**Certified training**: we choose auto-LiRPA Xu et al. (2020a) under CROWN-IBP + Loss Fusion setting. The learning rate is set to 0.001, and we use Adam with a weight decay of 0 for RSLoss and NRSLoss-based pruning and 0.00001 for other pruning methods. For the bound computation of NRSLoss and RSLoss, we use the bound produced by auto-LiRPA during certified training and use IBP during adversarial training.

---

[1]Note that for HYDRA pruning, the semi-supervised training scheme which exploits extra unlabeled data as in Sehwag et al. (2020) is NOT used in our experiments, for a fair comparison.

auto-LiRPA essentially uses IBP to compute bounds when input perturbation reaches a maximum during training and uses IBP constantly during testing. As we find the results to be unstable for certified training, we use five different random seeds, 100, 200, 300, 400 and 500 to initialize the training process, and then average the results of the same iterations. This is different from adversarial training where we only run one experiment.

We replace the standard IMP with weight rewinding Frankle & Carbin (2018) with each training method and each pruning method, and output the pruned subnetworks with different sparsity during iterative pruning and re-training. For each training method and each pruning method, we use one of the SOTA certified verifier Beta-CROWN Wang et al. (2021) to obtain the final accuracy of the subnetworks. For more experimental details, please refer to Appendix A.

### 4.1.4  Verifier and Evaluation Criterion

For each training method and each pruning method, we use one of the SOTA complete verifier Beta-CROWN Wang et al. (2021) to obtain the final accuracy of the subnetworks. Beta-CROWN is a highly GPU-parallelized verification framework and has SOTA performance in terms of bound tightness and verification speed. We choose ERAN benchmark Bak et al. (2021) which contains 1000 test images, and test on the first 200 images to reduce the time budget, since we note that there are no difference on average by comparing testing 1000 images and the first 200 images. We set the timeout of each test image to 300 seconds. We test the standard, adversarial, and verified accuracies, as well as time and GPU memory consumption of each model. We run the verifications using one NVIDIA RTX A6000 GPU card.

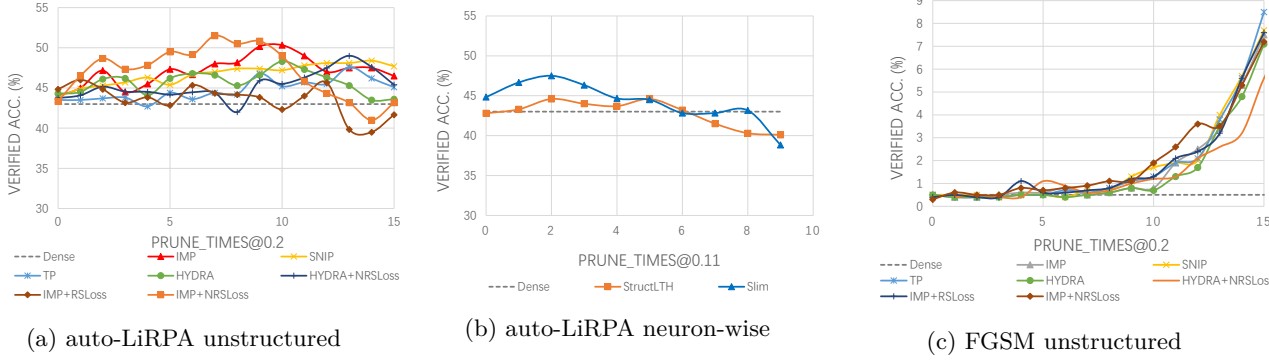

(a) auto-LiRPA unstructured  (b) auto-LiRPA neuron-wise  (c) FGSM unstructured

Figure 3: Verified Accuracy v.s. iterative pruning times (which can be converted to compression rate) on CIFAR10 dataset. (a) is unstructured pruning under auto-LiRPA training, (b) is neuron pruning under auto-LiRPA training (Note that *PRUNE_TIMES@0.11* means iterative pruning epochs with 0.11 **channel** pruning rate), (c) is unstructured pruning under FGSM. We move neuron pruning under FGSM to Figure 10 in Appendix due to page limit. Note that for FGSM setting, we use CROWN verifier instead of Beta-CROWN due to long verification time on Beta-CROWN. We observe signifcant improvement of verified accuracy for FGSM training mainly because pruning under adversarial training significantly trades verified accuracy with benign accuracy compared to pruning under certified training.

### 4.2  Experiment Results and Analysis

For the 3 benchmark datasets, the comprehensive experiment results under $\epsilon = 2/255$ are shown in Table 2, and the sample curves of Verified Accuracy v.s. iterative pruning times with seed 100 are shown in Figure 3. For each model, we verify all subnetworks produced by iterative pruning and report the best-verified accuracy and other corresponding evaluation metrics. We then choose several representative pruning methods that either have good performance or distinctive motivations or are important baselines, and evaluate them under different perturbation scales on the CIFAR10 dataset, as shown in Table 3. We also conduct a experiment to validate that weight-rewinding Frankle & Carbin (2018) is better than finetuning Sehwag et al. (2020) for improving certified robustness, as shown in Section B.2. The results of Random Pruning are omitted from

Table 2: Comparisons of subnetwork robustness and verification time for different training and pruning methods on FashionMNIST, SVHN, and CIFAR10 datasets. *std*, *adv*, *ver*, *t* refer to standard accuracy(%), adversarial accuracy(%), verified accuracy(%) and time consumption (s/sample), respectively. Note that the remain ratio refers to remaining weights for unstructured pruning, and remaining channels for neuron pruning, respectively. Note that for HYDRA pruning, we replace the adversarial loss in HYDRA with certified loss for Auto-LiRPA setting.

| | Training Method | | FGSM | | | | | auto-LiRPA | | |
| Dataset | Pruning Method | Remain Ratio | *std* | *adv* | *ver* | *t* | Remain Ratio | *std* | *ver* | *t* |
|---|---|---|---|---|---|---|---|---|---|---|
| | Dense | 1 | **85.2** | 81.2 | 1.5 | 298.3 | 1 | 77.2 | 68.8 | 7.23 |
| | IMP | 0.03 | 80.2 | 75.3 | 39.0 | 85.9 | 0.07 | 78.1 | 73.5 | 7.17 |
| | SNIP | 0.03 | 81.2 | 77.3 | 36.5 | 96.3 | 0.80 | 77.5 | 71.3 | 4.82 |
| Fashion-MNIST $\epsilon = 0.1$ | TP | 0.03 | 80.3 | 76.1 | 35.3 | 92.3 | 0.32 | 79.4 | 73.5 | 6.60 |
| | HYDRA | 0.03 | 81.5 | 77.2 | 33.5 | 105.9 | 0.51 | 79.1 | 73.3 | 6.60 |
| | HYDRA+NRS | 0.03 | 80.3 | 76.7 | 36.5 | 105.3 | 0.03 | 81.5 | 74.2 | 8.90 |
| | IMP+RS | 0.03 | 79.3 | 72.2 | 37.0 | 94.1 | 0.41 | **80.5** | 72.5 | 6.21 |
| | IMP+NRS | 0.03 | 81.2 | 74.9 | **41.5** | 78.4 | 0.05 | **80.5** | **74.0** | 6.06 |
| | StructLTH | 0.35 | 80.5 | 78.3 | 22.5 | 143.0 | 0.80 | 80.0 | 72.4 | 6.14 |
| | Slim | 0.35 | 80.7 | 78.3 | 31.0 | 105.0 | 0.32 | 78.5 | 71.8 | 2.96 |
| | Dense | 1 | **94.8** | 89.2 | 2.0 | 294.2 | 1 | 76.3 | 62.0 | 6.68 |
| | IMP | 0.03 | 92.3 | 84.4 | 31.0 | 188.3 | 0.16 | 82.0 | 67.3 | 6.65 |
| | SNIP | 0.03 | 92.4 | 84.3 | 29.1 | 204.4 | 0.32 | 82.5 | 66.5 | 5.93 |
| SVHN $\epsilon = 2/255$ | TP | 0.03 | 92.4 | 84.2 | 28.5 | 197.3 | 0.21 | 83.0 | 67.3 | 5.12 |
| | HYDRA | 0.03 | 91.2 | 82.7 | **41.5** | 146.3 | 0.64 | 79.5 | 66.3 | 8.75 |
| | HYDRA+NRS | 0.03 | 89.8 | 81.1 | 40.5 | 144.2 | 0.55 | 82.0 | 66.0 | 4.02 |
| | IMP+RS | 0.03 | 85.6 | 75.9 | 25.2 | 179.2 | 0.03 | 83.5 | 65.7 | 6.52 |
| | IMP+NRS | 0.03 | 92.1 | 83.2 | 33.5 | 153.2 | 0.08 | **86.0** | **68.3** | 6.06 |
| | StructLTH | 0.27 | 87.7 | 78.0 | 33.0 | 159.1 | 0.55 | 82.2 | 65.9 | 3.71 |
| | Slim | 0.35 | 89.7 | 78.7 | 35.5 | 151.8 | 0.63 | 84.0 | 65.3 | 7.77 |
| | Dense | 1 | **82.4** | 68.6 | 1.5 | 278.9 | 1 | 54.1 | 43.0 | 6.68 |
| | IMP | 0.03 | 62.2 | 55.4 | 23.5 | 135.3 | 0.13 | 61.0 | 50.1 | 6.65 |
| | SNIP | 0.03 | 61.5 | 55.1 | 22.5 | 128.4 | 0.04 | 59.8 | 48.4 | 6.67 |
| CIFAR10 $\epsilon = 2/255$ | TP | 0.03 | 59.7 | 55.4 | 24.0 | 132.4 | 0.05 | 59.9 | 47.6 | 6.02 |
| | HYDRA | 0.03 | 60.4 | 55.4 | 23.5 | 132.2 | 0.11 | 60.5 | 48.3 | 8.75 |
| | HYDRA+NRS | 0.03 | 54.9 | 48.4 | 25.0 | 132.2 | 0.05 | 58.0 | 49.0 | 8.75 |
| | IMP+RS | 0.03 | 60.2 | 54.2 | 23.5 | 134.4 | 0.13 | 58.6 | 46.3 | 6.52 |
| | IMP+NRS | 0.03 | 60.7 | 51.0 | 25.0 | 131.2 | 0.21 | **62.2** | **51.2** | 6.06 |
| | StructLTH | 0.35 | 55.6 | 48.3 | 14.0 | 143.7 | 0.55 | 57.5 | 44.6 | 3.71 |
| | Slim | 0.35 | 56.9 | 49.6 | **26.0** | 72.9 | 0.79 | 59.2 | 47.5 | 5.65 |

Table 2 and 3 since its standard accuracies are very poor and non-competitive (some can be found in Figure 3 (a) for illustration purpose). We also provide additional results of 2 different architectures using the same hyperparameter setting on CIFAR10 as shown in Appendix B.3, which shows that the relative performance of different pruning methods is stable across these 2 tested architectures. We next present the result analysis.

### 4.2.1 Can existing pruning methods improve certified robustness?

From Table 2 and Table 3, we observe general improvements in certified robustness brought by pruning, both under FGSM and auto-LiRPA settings. Specifically, Table 2 shows that under the auto-LiRPA setting, existing pruning methods can improve verified accuracies for $2.5 - 5.2\%$ on FashionMNIST, $3.3 - 6.3\%$ on SVHN, $1.6 - 7.1\%$ on CIFAR10, respectively and improve standard accuracies for $0.3 - 3.3\%$ on FashionMNIST, $3.2 - 9.7\%$ on SVHN, $3.4 - 8.1\%$ on CIFAR10 respectively, among which IMP consistently outperforms other existing pruning methods, with highest improvements of both standard and verified accuracies. This demonstrates that pruning can generally improve $L_\infty$ certified robustness. Moreover, under certified training, this improvement comes with no extra trade-off such as standard accuracy. We also observe that on more realistic datasets (SVHN, CIFAR10), the improvements under certified training are significantly bigger than that of the synthetic dataset (FashionMNIST).

Table 3: Verified accuracies of different pruning and robust training methods and perturbation scales $\epsilon$ under auto-LiRPA setting. Note that for HYDRA pruning, we replace the adversarial loss in HYDRA with certified loss.We also provide the performance curves with varying compression rate when $\epsilon=8/255$ in Figure 8 in Appendix. *std, ver, t* mean standard accuracy, verified accuracy and verification time, respectively.

| $\epsilon$ | | | 2/255 | | | | 8/255 | | |
|---|---|---|---|---|---|---|---|---|---|
| PRUNING TYPE | PRUNING METHOD | REMAIN RATIO | *std* | *ver* | *t* | REMAIN RATIO | *std* | *ver* | *t* |
| DENSE | | 1 | 54.1 | 43.0 | 6.68 | 1 | 36.1 | 29.3 | 5.83 |
| UNSTRUCTURED | IMP | 0.13 | 61.0 | 50.1 | 6.65 | 0.13 | 35.0 | 29.8 | 5.11 |
| | SNIP | 0.04 | 59.8 | 48.4 | 6.67 | 0.21 | 34.0 | 29.5 | 6.73 |
| | TP | 0.05 | 59.9 | 47.6 | 6.02 | 0.32 | 33.5 | 28.8 | 6.84 |
| | HYDRA | 0.11 | 60.4 | 48.3 | 8.75 | 0.11 | 34.5 | 29.5 | 6.36 |
| | HYDRA+NRS | 0.05 | 58.0 | 49.0 | 8.75 | 0.21 | 34.0 | 30.3 | 5.98 |
| | IMP+RS | 0.13 | 58.6 | 46.3 | 6.52 | 0.64 | 35.0 | 28.0 | 4.39 |
| | IMP+NRS | 0.21 | **62.2** | **51.2** | 6.06 | 0.51 | **39.0** | **31.5** | 4.32 |
| STRUCTURED | STRUCTLTH | 0.55 | 57.5 | 44.6 | 3.71 | 0.63 | 34.6 | 29.4 | 3.51 |
| | SLIM | 0.79 | 59.2 | 47.5 | 5.65 | 0.70 | 35.9 | 29.8 | 3.86 |

Under the FGSM setting, there are great improvements in verified accuracies with different pruning methods ranging from $21.0 - 40.0\%$ on FashionMNIST, $23.2 - 39.5\%$ on SVHN, and $12.5 - 24.5\%$ on CIFAR10, respectively, among which IMP+NRS, HYDRA and Network Slimming (Slim) obtain highest verified accuracy on FashionMNIST, SVHN, and CIFAR10, respectively. We also observe an obvious trade-off of standard/adversarial accuracy v.s. verified accuracy, i.e. with the big increase of verified accuracy after pruning, the standard and adversarial accuracies drop significantly. To explain this trade-off, we visualize the ratio of unstable neurons of different pruning methods, as shown in Figure 6 in Appendix. We find that the ratio of unstable neurons generally decreases as the sparsity gets higher, this is compliant with that neuron stability is important for certified robustness. However, if all neurons become stable, the whole network will become a linear function, which in turn withholds the standard accuracy. Hence the standard/verified accuracy trade-off is essentially the stability/expressiveness trade-off of the network. Nevertheless, this trade-off is not obvious under the auto-LiRPA setting since the training objective of auto-LiRPA incorporates standard accuracy.

Across different datasets, we observe general improvement brought by pruning for certified robustness, which consolidates our conclusion that pruning can generally improve $L_\infty$ certified robustness. In particular, NRSLoss-based pruning can outperform other pruning methods consistently under certified training and achieves competitive performance under adversarial training, which demonstrates the effectiveness of NRSLoss regularizer and the pruning scheme of IMP+NRS, as we explained in the methodology.

**Resource Consumption**: It can be observed from Table 2 and 3 that unstructured pruning tends to produce better performance than neuron pruning. However, neuron pruning has the advantage over unstructured pruning that it brings real hardware acceleration for certified verification, especially given that the computational overhead is a significant bottleneck for verifying large neural networks even with highly GPU-parallelized verifiers such as Beta-CROWN. We show an overview of the time and peak GPU memory consumption of neuron pruning under different pruning stages as in Figure 4. We can see that for every 3 prunings, which increases about 30% channel sparsity, the time consumption for models trained with auto-LiRPA can reduce by about 50%, whereas for models trained with FGSM can reduce by about $60 - 80\%$. We also observe that GPU memory consumption can be greatly reduced at high channel sparsity. The reduction in GPU memory consumption is even more important given the GPU memory bottleneck for complete verification of large neural networks. Furthermore, for these pruning methods, we observe similar high performing sparsity under different random seeds as shown in Figure 9 in Appendix, which means we do not need to verify every sparsity one by one to pick out the best sparsity, and it is crucial for accelerating the verification process in practice. We further discuss some bottleneck factors as we move towards verifying large networks with our introduced methods in Appendix C.

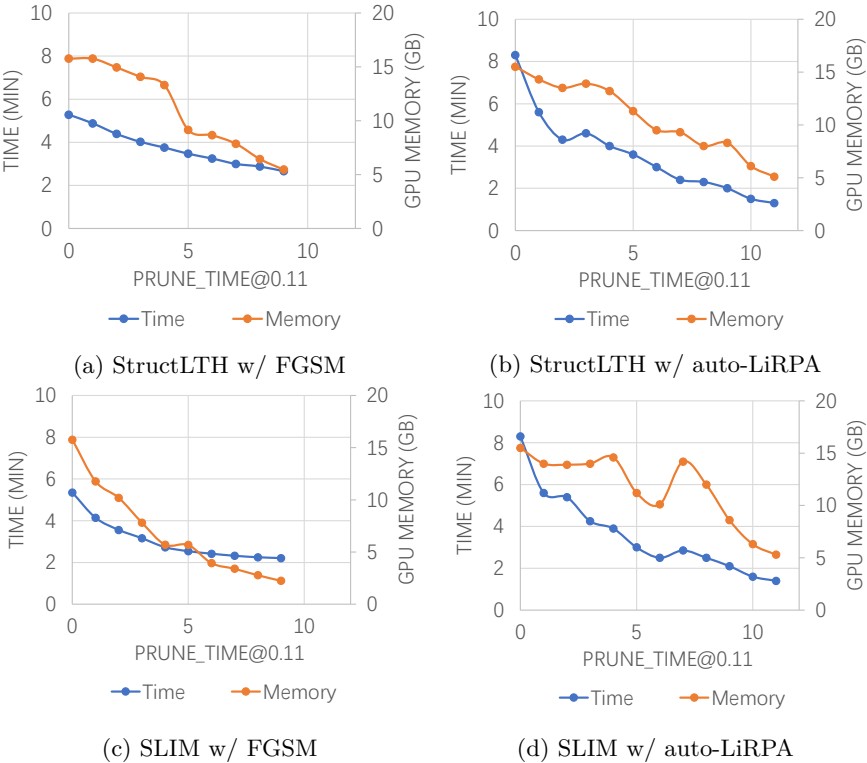

Figure 4: The mean verification time and average GPU memory consumption for neuron pruning methods on CIFAR10 dataset under CROWN mode. X-axis means the number of pruning iterations with 0.11 channel pruning rate. Note that we don't do such test on complete verification of Beta-CROWN mode which is time consuming but has highly similar trend to the results under CROWN mode.

### 4.2.2 How does NRSLoss-based pruning outperform other pruning methods?

From Table 2 and 3, we observe that IMP+NRS outperform other pruning methods under auto-LiRPA setting. Take the CIFAR10 dataset as an example, with 2/255 perturbation, IMP+NRS improves certified accuracy for 8.2% and standard accuracy for 8.1%; with 8/255 perturbation, IMP+NRS improves certified accuracy for 2.9% and standard accuracy for 2.3%. Notably, IMP+NRS achieves both the highest standard and verified accuracies, since the training objective of auto-LiRPA incorporates standard accuracy. By comparing HYDRA setting and HYDRA+NRSLoss setting, we observe NRSLoss can improve the verified accuracy for HYDRA pruning in FashionMNIST and CIFAR10 dataset, and has better standard/verified accuracy trade-off under certified training for SVHN dataset. We thus conclude that NRSLoss regularizer is effective for HYDRA pruning in most cases and is effective for IMP pruning for all cases we have tested. To demonstrate that the performance improvements of NRSLoss indeed come from stability-based regularization as discussed in Section 3, we visualize the pre-activation *network instability* (as proposed in Section 3) in Figure 5. We observe that the RS Loss and NRS Loss-based pruning have significantly lower instability compared to IMP, and the instability decreases as the sparsity gets higher, which proves that pruning with NRSLoss and RSLoss regularizer can decrease network instability, hence improving the certified robustness. It can also be observed that the RSLoss has lower instability than NRSLoss, however, since NRSLoss eliminates the gradients from BN layer, the RSLoss actually gets lower instability by influencing BN layers, which in turn would hurt normal training, and thus hurt overall performance. The advantage of NRSLoss can also be interpreted using the NRSLoss landscape as shown in Figure 2. Compared to RSLoss, NRSLoss takes account of the channel importance, so that using NRSLoss can avoid regularizing neurons that are in the important channels. From these results, we can again conclude that neuron stability is important for certified robustness, in particular, IMP+NRS motivated by improving neuron stability is effective for improving certified robustness.

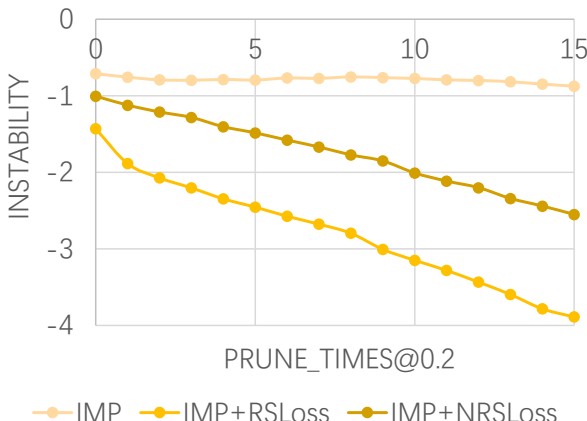

Figure 5: Network instability v.s. iterative pruning times of pre-activation and pre-BN forward pass values. The bounds are computed under 2/255 input perturbation using auto-LiRPA and the whole test set of CIFAR10. Note that we omit the curves of SNIP, TP and HYDRA for simplicity as they are almost coincident with the IMP curve.

We note that the literature results as in Wong et al. (2018) are better than the results reported in Table 2 and 3. In fact, it is achieved by using incomplete verifications on much larger neural networks (scaling the network size can improve the results), whereas we focus on the exploring the pruning effect for complete verifications because complete verifications give better (higher and exact) certified robustness results than incomplete verifications. However, existing complete verifiers are unable to verify these large networks as in Wong et al. (2018), due to OOM problem. For complete verification, our designed test network is the largest one that can be verified with complete verifiers on a GPU with 24GB memory.

### 4.2.3 Possible existence of certified lottery tickets.

As a possible finding, we demonstrate the possible existence of *certified lottery tickets* in our experiments, that generalizes the lottery ticket hypothesis Frankle & Carbin (2018) to certified robustness. Specifically, from Table 2, we observe that **all** pruning methods under certified training across all 3 datasets can find certified lottery tickets that can match *both* standard and verified accuracies to the original dense models, and most of the pruning methods can produce certified lottery tickets that significantly outperform original dense networks. From Table 3, we see that certified lottery tickets can be found on most pruning methods with a bigger perturbation scale except for random pruning and IMP+RS. From Figure 3(a), we observe that except for unstructured pruning (except for random pruning), certified lottery tickets occur almost in every sparsity. The above findings may reveal the possible existence of certified lottery tickets.

### 4.2.4 How should we choose pruning methods for certified robustness?

Generally, we would recommend IMP and IMP+NRS for performance concerns because they have the best verified accuracy under certified training across different datasets, and we recommend Network Slimming for efficiency concerns because its neuron pruning nature can essentially reduce the computational overhead of the complete verification. We empirically find that the relative performance of our tested pruning methods under certified training is similar to that under standard training. We conjecture that this is because an important goal of most pruning methods is causing a minimal negative influence on the training objective function, and this objective function is benign accuracy under standard training and verified accuracy under certified training, respectively. These pruning methods also implicitly regularize the network stability and bound tightness as stated in Section 3.3, which leads to general improvement compared to dense baselines. However, our proposed NRSLoss-based pruning explicitly regularizes network stability which makes it outperform other pruning methods.

### 4.3 Summary of Findings

In our experiments, we find that pruning can generally improve $L_\infty$ certified robustness for neural networks trained with different robust training methods and observe the possible existence of certified lottery tickets. Under adversarial training, we observe a significant trade-off between standard and verified accuracies with different pruning methods, but under certified training, pruning can improve both standard and verified accuracies. From our experiments, we know that RSLoss and NRSLoss are both effective at regularizing network stability but NRSLoss is better for imposing less regularization on more important neurons and removing the negative influence of stability regularization for BN layers. From Figure 4, we observe that neuron pruning can considerably reduce the computational overhead of complete verification for neural networks.

## 5 Conclusion

In this paper, we demonstrate that pruning can generally improve $L_\infty$ certified robustness, both for adversarial and certified training. We analyze some important factors that influence certified robustness, and offer a new angle to study the intriguing interaction between sparsity and robustness, i.e. interpreting the interaction of sparsity and certified robustness via neuron stability. In particular, we find neuron stability to be crucial for improving certified robustness, on which motivation we propose the novel NRSLoss-based pruning that outperforms existing pruning methods. We also observe the existence of certified lottery tickets. We believe our work has revealed the relationships between pruning and certified robustness, which can shed light on future research to design better sparse networks with certified robustness.

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

# Appendix

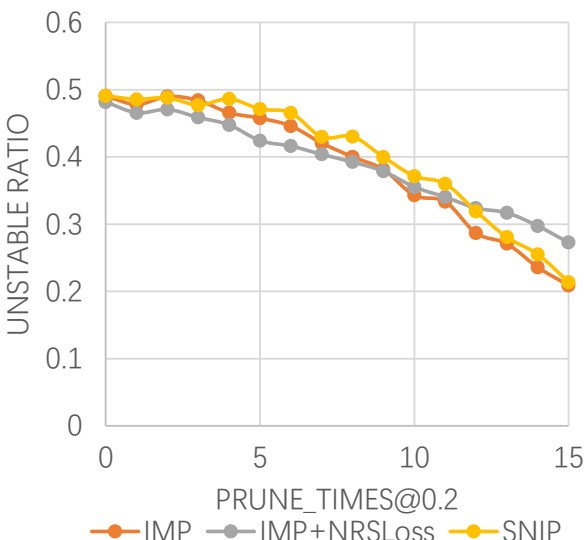

Figure 6: The ratio of unstable neurons v.s. pruning times of different pruning methods under FGSM setting. We empirically find that all pruning methods have similar curves in this figure, so we sample 3 curves to demonstrate for simplicity. Note that *PRUNE_TIMES@0.2* means the iterative pruning epochs with 0.2 pruning rate, and we reuse this notation hereafter.

## A  Experiment details

FGSM and auto-LiRPA share some common hyperparameters. The batch size is set to 128, and we clip the norm of gradients to a maximum of 8. We train 200 epochs in one pruning iteration for each experiment, with a learning rate decay factor of 0.1 at 140 and 170 epochs. We set the input perturbation $\epsilon$ as $L_{inf}$-norm ball to 0.1 for the FashionMNIST dataset and 2/255 for SVHN and CIFAR10 datasets, and gradually increase $\epsilon$ from 0 to 2/255 starting from 11th epoch and until 80th epoch. We also scale the perturbation to 8/255 to validate the effectiveness of pruning under bigger perturbations. The training epochs under 8/255 perturbation is set to 300. After each pruning iteration, we rewind the remaining weights to initial states and reset the optimizer with the initial learning rate and $\epsilon$.

## B  Ablation

In this section, we conduct 2 ablation studies mainly on the CIFAR10 dataset to further consolidate our claims.

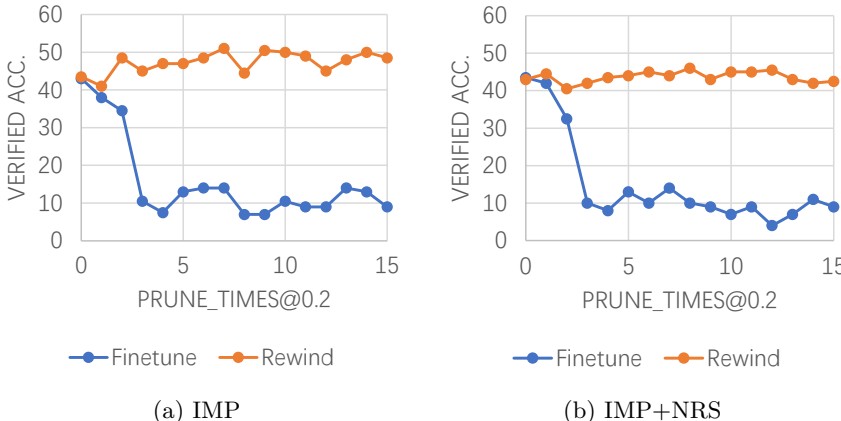

(a) IMP  (b) IMP+NRS

Figure 7: Comparison of finetuning and weight-rewinding verified accuracy v.s. pruning times under auto-LiRPA setting.

Table 4: Comparison of auto-LiRPA and Fast-IBP on CIFAR10 dataset. Note that for HYDRA pruning, we replace the adversarial loss in HYDRA with certified loss.

| Training Method | | | auto-LiRPA | | | | FastIBP | | |
|---|---|---|---|---|---|---|---|---|---|
| Pruning type | Pruning Method | Remain Ratio | $std$ | $ver$ | $t$ | Remain Ratio | $std$ | $ver$ | $t$ |
| Dense | | 1 | 54.1 | 43.0 | 6.68 | 1 | 56.0 | 43.5 | 8.42 |
| | IMP | 0.13 | 61.0 | 50.1 | 6.65 | 0.04 | 63.5 | 53.0 | 6.17 |
| | HYDRA | 0.11 | 60.5 | 48.3 | 8.75 | 0.13 | 64.0 | 51.0 | 6.56 |
| | IMP+RS | 0.13 | 58.6 | 46.3 | 6.52 | 0.17 | 61.5 | 47.0 | 5.98 |
| | IMP+NRS | 0.21 | 62.2 | 51.2 | 6.06 | 0.17 | 64.0 | 54.0 | 5.94 |
| Structured | Slim | 0.79 | 59.2 | 47.5 | 5.65 | 0.44 | 56.0 | 46.5 | 5.53 |

### B.1  Comparison of pruning under different certified training methods

In our main experiments, we choose the auto-LiRPA as the certified training method. The reason we choose this method is based on its training efficiency and competitive performance, and its training efficiency mainly comes from the *loss fusion* technique as proposed in Xu et al. (2020a). The training efficiency is important in our experiments because we use iterative training and pruning, which boosts the overall training time

to 16 times longer. We notice that the certified training methodShi et al. (2021) (denoted as **FastIBP** in the following context) with SOTA performance (i.e. SOTA verified accuracy) claims that loss-fusion has a negative influence on the performance and thus doesn't adopt it in their method. We empirically find that without loss-fusion, the training speed of FastIBP is 4 times slower than auto-LiRPA. We thus choose auto-LiRPA as the certified training method in our main experiments. However, we here present a comparison of results (see Table 4) of these 2 certified training methods on the CIFAR10 dataset and pruned with several pruning methods, to demonstrate that the improvement of certified robustness brought by pruning is consistent with different certified training methods. The hyperparameter settings are the same as mentioned in our main experiments. From Table 4, we observe better performance can be obtained with FastIBP, and standard/verified accuracies are consistently improved with different pruning methods, among which IMP+NRS still achieves the best performance.

## B.2 Comparison of finetuning and weight-rewinding for pruning

We empirically find that after each pruning, rewinding the network parameters to their initial states as in Frankle & Carbin (2018) produces better performance than finetuning the parameters as in Sehwag et al. (2020). Specifically, we follow the experiment setup as in Section 4.1, except that for finetuning mode we don't re-initialize the learning rate after pruning. The results are demonstrated in Figure 7. We observe that the finetuning-based pruning always produces worse performance than weight rewinding-based pruning, and its accuracy tends to collapse at 3rd pruning iteration. Therefore we conclude that weight-rewinding-based pruning is more effective than finetuning-based pruning for certified robustness.

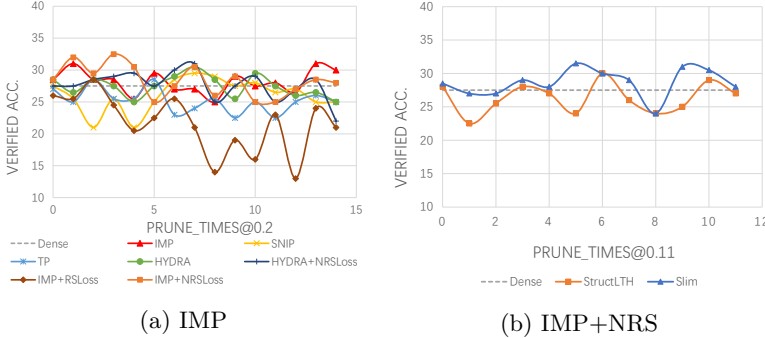

(a) IMP       (b) IMP+NRS

Figure 8: Verified Accuracy v.s. iterative pruning times with 8/255 perturbation on CIFAR10 dataset. (a) is unstructured pruning under auto-LiRPA training with 0.2 element-wise pruning rate, (b) is neuron pruning under auto-LiRPA training with 0.11 channel-wise pruning rate.

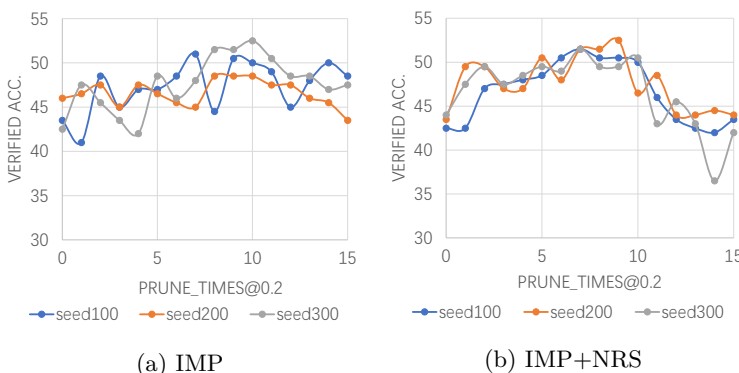

(a) IMP       (b) IMP+NRS

Figure 9: Verified accuracy v.s. iterative pruning times under different random seeds with auto-LiRPA training. (a) is IMP pruning and (b) is IMP+NRS pruning.

Table 5: Comparison of CNN-7L and ResNet-18-NR on CIFAR10 dataset.

| Architecture | | | CNN-7L | | | | ResNet-18-NR | | |
|---|---|---|---|---|---|---|---|---|---|
| Pruning type | Pruning Method | Remain Ratio | *std* | *ver* | *t* | Remain Ratio | *std* | *ver* | *t* |
| Dense | | 1 | 54.1 | 43.0 | 6.68 | 1 | 53.5 | 42.4 | 5.31 |
| Unstructured | IMP | 0.13 | 61.0 | 50.1 | 6.65 | 0.21 | 59.5 | 49.1 | 5.67 |
| | SNIP | 0.04 | 59.8 | 48.4 | 6.67 | 0.17 | 57.2 | 47.4 | 5.21 |
| | TP | 0.05 | 59.9 | 47.6 | 6.02 | 0.05 | 56.8 | 46.2 | 6.18 |
| | HYDRA | 0.11 | 60.4 | 48.3 | 8.75 | 0.13 | 58.7 | 47.0 | 5.96 |
| | HYDRA+NRSLoss | 0.05 | 58.0 | 49.0 | 8.75 | 0.05 | 57.2 | 47.5 | 6.12 |
| | IMP+RS | 0.13 | 58.6 | 46.3 | 6.52 | 0.44 | 56.2 | 46.0 | 6.87 |
| | IMP+NRS | 0.21 | **62.2** | **51.2** | 6.06 | 0.11 | **60.3** | **49.6** | 5.92 |
| Structured | StructLTH | 0.55 | 57.5 | 44.6 | 3.71 | 0.70 | 56.2 | 43.4 | 4.10 |
| | Slim | 0.79 | 59.2 | 47.5 | 5.65 | 0.79 | 58.6 | 46.0 | 4.54 |

### B.3 Comparision of different architecture

We provide supplemental results on comparing the performance of 2 different architectures using the introduced methods on CIFAR10 dataset. Besides the 7-layer feed-forward CNN (namely CNN-7L) we adopt in the main experiments, we also test the recently provided ResNet-18 network (narrowed version) as in the official alpha-beta-CROWN repo[2], and double its network width which leads to 4 times of parameter number. Due to the fact that network depth has more influence on GPU memory occupation, our modified version of ResNet-18 (namely ResNet-18-NR) has the biggest width that can be fitted in 24GB GPU memory, despite that it has smaller parameter number than CNN. We follow the same hyperparmeter setting as in Table 2, and the results are shown in Table 5. We observe that the relative performance is stable across different pruning methods when we change the network architecture, and IMP+NRS pruning still achieves the best performance.

## C will the size of the network or dataset hinder the introduced pruning and verification process?

SOTA complete verification methods such as the Alpha-beta-CROWN we use in this paper have exponential GPU memory consumption due to the BaB framework, so as the network size grows (especially when growing deeper), the complete verification would be harder to be completed due to the hardware constraint. Larger dataset typically requires larger networks to achieve good performance, so the verification would also be indirectly affected by dataset size. This is a common problem faced by vanilla certified training. However, for subnetworks obtained by neuron pruning during certified training, this problem can be relieved (See Figure 5 for example). For training and inference efficiency, our proposed regularizer has the same complexity with a forward pass of the network since we use IBP to compute the neuron bounds, therefore the training and inference stage of the model with our proposed regularizer won't be affected by the network or dataset size.

## D Is pruning effective for certified robustness at scale?

In light of exploring whether pruning would be effective for certified robustness at scale, we conduct experiments on the downscaled 64x64 ImageNet Chrabaszcz et al. (2017), following the experiment setting in Table 4 from Xu et al. (2020a). Specifically, there are 1,000 class labels and the $L_\infty$ input perturbation scale is set to 1/255. We use WideResNet trained with auto-LiRPA and iterative pruning, and also verify using auto-LiRPA incomplete verifier since complete verifiers have OOM issue on WideResNet. The results are shown in Table 6. We observe consistent improvements across different pruning methods even when is verified accuracies are

---

[2]https://github.com/Verified-Intelligence/alpha-beta-CROWN/blob/main/complete_verifier/exp_configs/cifar_resnet18.yaml

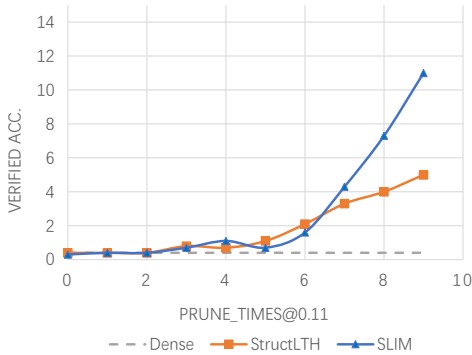

Figure 10: Verified Accuracy v.s. iterative pruning times on CIFAR10 dataset of neuron pruning under FGSM training.

Table 6: The incomplete verification results on downscale ImageNet, following the experiment seeing in Table 4 from Xu et al. (2020a). The perturbation $\epsilon$ is 1/255 and the tested network is WideResNet. The model is trained and (incompletely) verified with Auto-LiRPA. *std, adv, ver* refer to standard accuracy, adversarial accuracy obtained with PGD attack, and verified accuracy .

|  | Dense | IMP | TP | SNIP | HYDRA | HYDRA +NRS | IMP +RS | IMP+ NRS | Struct -LTH | Slim |
|---|---|---|---|---|---|---|---|---|---|---|
| Remain Ratio | 1 | 0.21 | 0.51 | 0.32 | 0.32 | 0.21 | 0.40 | 0.13 | 0.55 | 0.35 |
| *std* | 83.77 | 86.18 | 85.57 | 86.30 | 85.78 | 85.96 | 84.12 | 86.32 | 84.76 | 85.48 |
| *adv* | 81.74 | 84.23 | 83.89 | 84.14 | 83.67 | 83.53 | 83.10 | 84.64 | 82.90 | 83.74 |
| *ver* | 91.27 | 93.22 | 92.43 | 92.66 | 92.35 | 92.75 | 92.07 | 93.80 | 91.43 | 91.27 |

already high. Our proposed NRSLoss-based pruning still remains competitive among all pruning methods we tested.

