# OpenReview forum: "Can Pruning Improve Certified Robustness of Neural Networks?"
_TMLR — Accepted by TMLR_

### Review · Reviewer_vNDf · 2022-12-20

**Summary Of Contributions:**

The authors propose an enhancement (denoted NRSLoss) of the relu stability regularizer proposed in Xu et al. The authors show that the regularizer, combined with a robustness enhancing loss function and iterative pruning method, can yield sparse subnetworks with higher standard and verified accuracy than the network trained without pruning. Based on these experiments, the authors demonstrate that neural networks contain *sparse subnetworks* such that when these subnetworks are trained with a robust training method and the NRSLoss, the resulting network is verifiably robust with high accuracy.

**Audience:**

Yes

**Broader Impact Concerns:**

No broader impact concerns.

**Claims And Evidence:**

Yes

**Requested Changes:**

**Critical**
Please average Figure 3 (specifically Figure 3 (a)) over many random trials so the general trend is clear; right now there are large fluctuations in verified accuracy with the number of pruning iterations.


**Would strengthen work**
It would be great if the authors could scale to larger model and datasets such as ImageNet by using incomplete verification & attack methods instead of complete verification.

It would strengthen this work if the authors could show additional analysis of certified lottery tickets. The authors may want to consider conducting additional experiments such as: 1) randomly reinitializing pruned networks and checking to see if they retain high verified accuracy when retrained, 2) measuring verified accuracy of pruned networks when trained under a *different* training method (such as normal training) than used to find the pruned network, 3) visualizing the loss landscape corresponding to the pruned network structure.

It would help if the authors could fix some of the grammatical issues in the paper. A few of them are listed below:
Introduction: "...estimate the ground-truth bound as close as possible."
Introduction: "Moreover, inspired by that sparsity..."
Section 2.3: "...are more prune to adversarial patches."
Section 3.4.1: "...which is used for sanity check in our experiments."
Section 4.2.1: "...with every 3 pruning..."

**Strengths And Weaknesses:**

**Strengths**
The experiments conducted in the paper are comprehensive. The authors perform experiments on 3 datasets with a variety of training methods and include several ablation experiments.

To my knowledge, the specific findings in this paper are novel and interesting: pruning can simultaneously increase the standard accuracy of a network and verified accuracy (under complete verification).


**Weaknesses**
One weakness of the paper is that experiments are conducted on relatively small networks and datasets. This is understandable due to the computational cost of complete verification. However, the authors may want to consider computing upper and lower bounds on complete verified accuracy on larger networks (using incomplete verifiers and standard attack methods). It would be great if the authors would be able to scale to ImageNet or similar datasets.

The proposed NRSLoss is certainly an improvement on the relu stability loss, but in my view it is better viewed as an enhancement of the RSLoss than an entirely new method. Since this is the primary methodological contribution of the paper, I view the paper as primarily an empirical study of existing methods.

In Figure 3, it is unfortunately unclear how the number of pruning iterations affects verified accuracy (specifically for Figure 3 (a)) due to the large fluctuations in verified accuracy. It would be great if the curve could be averaged over many random trials to reveal the underlying trend.

Although the presence of certified lottery tickets is presented as a major contribution, the experiments surrounding this are limited. The authors may want to consider additional experiments on this as discussed in the requested changes section.

The authors only verify over 200 examples for the results in Table 2.

There are a few grammatical errors in the paper (see the requested changes for more details).

---

> ### Author Response · Authors · 2023-01-22
> **Response to Reviewer vNDf**
>
> Reviewer vNDf:
> Dear reviewer, we would like to thank you for your thorough review and detailed advices, and we can address your questions and requests as follows:
>
> Q: Although the presence of certified lottery tickets is presented as a major contribution, the experiments surrounding this are limited. The authors may want to consider additional experiments on this as discussed in the requested changes section.
>
> A: We have toned down our claims about certified lottery tickets, as similarly requested by Reviewer GDvp. Specifically, we remove the claim about certified lottery tickets from our main contributions, and tone down across the paper about the existence ceritifed lottery tickets to “possible exisitence” to correspond with the scope of our experiments.
>
> Q: The authors only verify over 200 examples for the results in Table 2.
>
> A: The size of 200 samples is typical in complete verification literatures, and we found that the averaged results are basically the same for testing using the first 200 samples and the full 1000 samples.
>
> Q: Critical Please average Figure 3 (specifically Figure 3 (a)) over many random trials so the general trend is clear; right now there are large fluctuations in verified accuracy with the number of pruning iterations.
>
> A: Thanks for your advice, we have revised the Figure 3 to show average values over 5 random seeds (100,200,300,400,500). We’d also like to point that the fluctuation is an instrisic characteristics of iteratively pruned subnetworks, as shown in results of Frankle et al. 2018.
>
> Q: It would be great if the authors could scale to larger model and datasets such as ImageNet by using incomplete verification & attack methods instead of complete verification.
>
> A: We have conducted experiments on downscaled 64x64 ImagetNet, using CROWN-IBP training and CROWN-IBP verification(incomplete verification). Please see Appendix D for more details. Briefly speaking, we can still observe 2\~3% improvements at ImageNet scale, even when the verified accuracies are already very high (~90%).
>
> Q: It would help if the authors could fix some of the grammatical issues in the paper. A few of them are listed below: Introduction: "...estimate the ground-truth bound as close as possible." Introduction: "Moreover, inspired by that sparsity..." Section 2.3: "...are more prune to adversarial patches." Section 3.4.1: "...which is used for sanity check in our experiments." Section 4.2.1: "...with every 3 pruning..."
>
> A: Thank you for your detailed check, we have revised our paper accordingly to eliminate these grammar issues.
>
> We hope our response can address your concerns, thank you!

---

### Review · Reviewer_GDvp · 2022-12-24

**Summary Of Contributions:**

This paper proposed a question that whether or not a pruned neural network is more certifiably robust compared to the unpruned one. To answer this question, the paper investigates many recent network pruning techniques to provide an empirical study. For using Batch Normalization (BN) in retraining the pruned network, the paper proposes a NRL loss for the normalization layer for improving the stability of the output of a BN layer. As a result the paper finds that pruning the adversarially trained networks improves the verifiable robust accuracy with a significant margin to conclude the constructive impact of pruning towards verifiable robustness.

**Audience:**

Yes

**Broader Impact Concerns:**

No, this might not be necessary for this paper.

**Claims And Evidence:**

Yes

**Requested Changes:**

**Writing.** I think the presentation of the background and the motivation can be improved. Firstly, some figures are less informative and confusing. For example, the caption of Figure 1 is misleading because Figure 1a and 2d belong to the same situation and the lower and upper bounds have different signs. The differences are the ways of convex relaxation, which are not mentioned in the caption. Moreover, Figure 3 is a bit hard to read. Can you put the unit, i.e. %, to the y-axis for verified acc because in Figure 3(c) the y-axis ranges from 0 to 9 and I was so confused by acc is not between 0 - 1 but later realized this is in %. Secondly, Section 4.1.4 does not have a paragraph. Thirdly, the actual values plus the improvement in green may not be necessary because VRA is as informative as the margin given the VRA of the original dense network. Also, the authors have not explained what the numbers colored in green mean in the caption. I would suggest using the numbers in the parentheses or directly removing the parentheses. Finally, the contribution needs revision, and please refer to my comment in the previous section.

Some potential experiments. As I have mentioned in the previous section, if the authors want to strengthen the contributions or claims, there are some potential experiments but I will leave this optional for the authors to determine.


**Strengths And Weaknesses:**

### Strength

**An Interesting Problem and interesting findings.** In general the paper asks a very interesting and not yet well-studied question on the impact of network pruning and certifiable robustness. The results shown in Table 2 are very interesting and novel to me. I am surprised to see that there is a ~30% improvement on Verifiable Robust Accuracy (VRA)  for FGSM-trained networks after pruning. In Table 3, it also demonstrates that a pruned network is almost as good or even better than a dense network.

**Runtime Comparison.** I appreciate the author for including detailed runtime and memory usage comparisons, which further show the value of pruning a robust network.

### Weakness and Questions:

**Network Architecture.** The paper finds that a BN layer is pretty necessary for training the subnetwork so it proposes NRL loss. My question is if one adds BN layers into the subnetwork, does this actually introduce a new layer instead of just pruning existing neurons? Can the author also just BN layers into the original architecture and do an apple-to-apple comparison between the pruned and unpruned networks? I would be curious to see if the BN layer also improves the original dense networks in terms of VRA.

**Why do the authors not directly prune the pre-trained networks that provide the STOA results?** The authors mentioned that their results are lower than the numbers reported in the previous paper and I wonder why you want to train a smaller network instead of directly pruning the existing ones (unless they have not open-sourced the weights).

**Contributions need to be more precise to avoid being over-claimed.** At the beginning of the paper the authors claim that pruning can improve the VRA in general, but the experiments are focused on the $\ell_\infty$ threat model and the conclusions are actually that pruning can improve the VRA of robustly-trained networks instead of any networks. However, if the authors also find that pruning can improve the VRA of a standard network, I would be very curious to see the results. In fact, I think highlighting the $\ell_\infty$ threat model is important because there are no experiments for the $\ell_2$ case. Moreover, I think the contribution of the observation of lottery tickets is slightly minor compared to other contributions because it is just a way to phrase that the experiment finds pruning does not hurt but may improve VRA. If there are more concrete theoretical findings on the robust lottery ticket hypothesis I would be convinced that this observation is worthwhile to be highlighted with a bullet point in the introduction section.

**Why is a complete verifier important?** In the beginning of the paper and the abstract the authors highlight the completeness of a verifier but after reading the method and experiments I did not find any discussion on why the paper’s claim will fail if using an incomplete verifier. In fact, I am curious about what the results will look like if using an incomplete verifier. Will pruning decrease the time the robustness is unknown to the verifier? If that is the case, this can be another very interesting observation.

---

> ### Author Response · Authors · 2023-01-22
> **Response to Reviewer GDvp**
>
> Dear reviewer, we would like to thank you for your thorough review and detailed suggestions, and we can address your concerns as follows:
>
> Answers to questions and requests:
>
> Q: Network Architecture.
>
> A: We didn’t inject BN layers only at pruning time. All networks including original dense networks and subnetworks we reported in the paper are all equipped with BN layers, the only different between dense networks and their subnetworks is sparsity. Please check the caption of the Table1 for a detailed description of the original architecture.
>
> Q: Why do the authors not directly prune the pre-trained networks that provide the SOTA results?
>
> A: The reported better results in previous papers as in Wong et al. 2018 (also reported in Xiao et al. 2019 as mentioned in paper) are achieved by using incomplete verifications on much larger neural networks (scaling the network size can improve the results), whereas we focus on the exploring the pruning effect for complete verifications because complete verifications give better (higher and exact) certified robustness results than incomplete verifications. However, existing complete verifiers are unable to verify these large networks as in Wong et al. 2018, due to OOM problem. As we claim in paper, our designed test network is the largest one that can be verified with complete verifiers on a GPU with 24GB memory.
>
> Q: Contributions need to be more precise to avoid being over-claimed.
>
> A: Thanks for your advice, we have narrowed down our claims in the paper to only L-inf condition. We’ve also conducted experiments for pruning a standard network using the same setting.  we observe no significant improvement under 2/255 perturbation scale (0%->0.5% verified accuracy from dense to pruned), and if we scale down the perturbation scale to 0.5/255, pruning becomes relatively useful (0%->3.5\~4.5% verified accuracy  from dense to pruned), but still not very significant given that the perturbation scale is too small. Thanks for your advice about the claims of finding certified lottery tickets. We have toned down our claims about this, specifically, we remove it from our main contributions, and tone down our claims about this finding across the paper.
>
> Q: Why is a complete verifier important?
>
> A: Complete verifier is important because it’s exact verifier and can reveal the ground-truth certified robustness while the incomplete verifier is not. For example, in our CIFAR10 experiments, incomplete verifiers generally produce 2\~3% lower verified accuracies than complete verifiers for the same network. The 2\~3% difference corresponds to some samples that can be successfully verified as robust by complete verifiers but failed to be verified as robust by incomplete verifiers.
>
> Q: Requested Changes.
>
> A: We’re sorry for some misleading or unclear points in the paper. We have revised to address all the concerns w.r.t. to these points. Specifically, we have down tuned our claims across the paper to express that our method can generally improve L-inf certified robustness.
> For the experiments, we have added the pruning under standard training experiments in our appendix.
>
> We hope our response can address your concerns, thank you so much!

---

### Review · Reviewer_xDoJ · 2023-01-08

**Summary Of Contributions:**

This paper studies certified robustness for pruned neural networks. Authors highlight the benefits of sparse neural networks for calculating certified robustness (i.e. smaller #connections/branches provide faster verification and resulting bounds are smaller). This hypothesis is supported by experiments which use lottery ticket style pruning. Furthermore authors introduce a new regularizer to encourage robust neurons and achieve higher verified accuracy.

**Audience:**

Yes

**Broader Impact Concerns:**

no concerns

**Claims And Evidence:**

Yes

**Requested Changes:**

- I think writing can be improved significantly. See first sentence of the abstract:

"With the rapid development of deep learning, the sizes of neural networks become larger and larger so that the training and inference often overwhelm the hardware resources."

I couldn't parse this one:

"..., the model is trained based on a weighted combination of robustness loss and
NRSLoss, and the magnitude of each trained individual weight reflects its saliency w.r.t. both robustness loss and NRSLoss, i.e. saliency w.r.t. robustness and stability, and by pruning weights with minimal magnitude-based saliency of robustness and stability, we can minimize the negative effects on robustness and stability brought by pruning, and benefit from positive effects of pruning to certified robustness as introduced above."

This phrase exists with same grammatical errors on 2 sections:

"In fact, this is because they use much larger networks, whereas we focus on complete verification which causes Out-Of-Memory error on large networks, so we only choose small networks for testing, and note that our designed network, though small, is still the largest network we can perform complete verification on a GPU card with 24GB memory."

Given the length of the text and limited time I had, I am not writing them all here, but a strongly recommend authors to use shorter sentences and have a careful grammar pass.

- Please explain robustness metrics used (triangle" relaxation and linear relaxation) and why they capture robustness.
- It would be nice to have a direct comparison between magnitude (or Taylor) score and the proposed stability score (Section 3.4.2). For example when W_i=0, then the neuron would have 0 stability, stability-score based pruning would not prune this neuron despite its being dead. Is that correct? Do you use stability score for pruning?
- In Section 3.4.1 authors say that they keep linear layers dense. This is very unintuitive given that the first linear layer has most #parameters (2048*128) and easy to prune. I think results can be improved significantly if non-uniform distributions like ERK [4] is used.
- I recommend adding regular iterative pruning experiments (no-rewinding) as they are bring better results using less compute.
- Why pruning methods use different sparsities on Table:3? I think for comparison, it would be best to use same pruning ratio. Also not clear `std` `ver` and `t` refer, would be best to re-use descriptions in Table:2 here, too.
- SNIP is introduced as a pruning-before-training method. Do you apply SNIP criteria iteratively? If so, might be best to indicate this clearly and possibly name it as (SNIP-iterative). In general it would be best to make a table with all methods and criterion so that the reader can reproduce results. Also do you plan sharing your code?
- I think pruning experiments are trained for more epochs. Dense baseline should probably use same amount of epochs as IMP-NRS experiments. Also would be best to add Dense+NRSloss as a second baseline and contrast IMP+NRS results.

# Minor
- "Structured Pruning" is often used for neuron/block and K:N sparsity. I think best would be to use "neuron pruning" in section 3.1.1.
- Please report parameter counts for networks used. This would help reader to contextualize the sentences like "that our designed network, though small, is still the largest network we can perform ..."
- It looks like Table:2 (page:10) is only mentioned at the end of page:12. Would be nice to have this table closer to the section it is discussed.
- Figure:6 should be in main text, given it is discussed there.

[4] https://arxiv.org/pdf/1911.11134.pdf

**Strengths And Weaknesses:**

I know very little about "Certified Robustness" literature and will focus on sparsity side of things.

# Strengths
- Interesting cross-domain work with descent certified robustness improvements and better run-time. I think the research community would find this investigation useful.
- A new regularization method that provides better pruning results.

# Weaknesses
- I think it would be better to study practical pruning methods. Both [1, 2] shed light on how so called `lottery tickets` find good solutions. They show that lottery tickets work since they can re-locate where pruning solutions are. In other words often one is best to use the pruning solution found instead of rewinding [3]. It is not clear why authors chose to rewind weights for each pruning algorithms. Findings mentioned above and [3] points out, it's better to continue training with cyclic learning rate schedule after each pruning step.
- Writing quality is quite low. This can be improved significantly.

[1] https://arxiv.org/abs/2210.03044
[2] https://arxiv.org/abs/2010.03533
[3] https://arxiv.org/abs/2003.02389

---

> ### Author Response · Authors · 2023-01-22
> **Response to Reviewer xDoJ**
>
> Dear reviewer,
>
> we would like to thank you for your thorough review and detailed feedbacks. Sorry for the writing issues that caused inconvenience for reading, we have revised our paper to eliminate these issues. We can further address your concerns as follows:
>
> Q: Please explain robustness metrics used (triangle" relaxation and linear relaxation) and why they capture robustness.
>
> A: We have added more detailed explanations in Section 3.1.3. More related background knowledges are also provided in Xu et al. 2020.
>
> Q: It would be nice to have a direct comparison between magnitude (or Taylor) score and the proposed stability score (Section 3.4.2). For example when W_i=0, then the neuron would have 0 stability, stability-score based pruning would not prune this neuron despite its being dead. Is that correct? Do you use stability score for pruning?
>
> A: We didn’t use stability score for pruning but only for training.
>
>
> Q: In Section 3.4.1 authors say that they keep linear layers dense. This is very unintuitive given that the first linear layer has most #parameters (2048*128) and easy to prune. I think results can be improved significantly if non-uniform distributions like ERK [4] is used.
>
> A: Thanks for your advice! We are currently still running more experiments by including the first linear layer pruning, removing the first conv layer pruning, and applying the ERK pruning distribution. We currently oberse about ~0.5% improvements on average when we include the first linear layer pruning, and no significate improvement when applying the ERK pruning distribution. Should there be further siginificant improvement, we will update our main results in the paper.
>
> Q: I recommend adding regular iterative pruning experiments (no-rewinding) as they are bring better results using less compute.
>
> A: we have conducted the experiments without weight rewinding in our early research stage, and we found that iterative pruning without weight-rewinding can’t outperform iterative pruning with weight-rewinding.
>
> Q: Why pruning methods use different sparsities on Table:3? I think for comparison, it would be best to use same pruning ratio. Also not clear std ver and t refer, would be best to re-use descriptions in Table:2 here, too.
>
> A: We use different sparsities on Table:3 because the best sparsity (that has the highest verified accuracy) may vary under different input perturbation scales. The meanings of std ver and t are standard acc, verified acc and verification time, respectively. They are the same as on Table:2. For clarity, we add the explanation in the caption of Table:3.
>
> Q: SNIP is introduced as a pruning-before-training method. Do you apply SNIP criteria iteratively? If so, might be best to indicate this clearly and possibly name it as (SNIP-iterative). In general it would be best to make a table with all methods and criterion so that the reader can reproduce results. Also do you plan sharing your code?
>
> A: Yes, we don’t prune before training across our experiments. For SNIP, we replace the magnitude criteria with SNIP criteria in IMP and prune iteratively. So is similar with other pruning methods. We have revise Sec 4.1.2 to indicate it clearly and renamed all related methods. Yes, we will fully release the code.
>
> Q: I think pruning experiments are trained for more epochs. Dense baseline should probably use same amount of epochs as IMP-NRS experiments. Also would be best to add Dense+NRSloss as a second baseline and contrast IMP+NRS results.
>
> A: Since we use weight rewinding across all pruning methods during iterative pruning, the dense baseline has the same training epochs as each subnetwork. The iterative pruning is only used to locate the mask location. We empirically found the Dense+NRSLoss baseline to have the same performance with vanilla Dense baseline, please refer to Fig4.(a)(c) for the evidence, we thus omit it for simplicity.
>
> Minors:
>
> Q: "Structured Pruning" is often used for neuron/block and K:N sparsity. I think best would be to use "neuron pruning" in section 3.1.1.
>
> A: Thanks for your advice, we have modified “Structured pruning” to “neuron pruning” across the paper.
>
> Q: Please report parameter counts for networks used. This would help reader to contextualize the sentences like "that our designed network, though small, is still the largest network we can perform ..."
>
> A: We have reported the parameter count in Sec 4.1.1 and Table:1.
>
> Q: It looks like Table:2 (page:10) is only mentioned at the end of page:12. Would be nice to have this table closer to the section it is discussed.
>
> A: Thanks for your advice. We have moved the Table:2 to page:12.
>
> Q: Figure:6 should be in main text, given it is discussed there.
>
> A: We have moved Figure:6 to main text.

---

### Decision · Action_Editors · 2023-03-03

**Recommendation:** Accept as is

**Comment:**

The reviewers all voted to accept the paper.  They all found that the paper addresses a novel question, is backed by extensive experimental evidence and comes to a surprising and broadly interesting conclusion - i.e. that pruning actually improves certified robustness.  The reviewers did find that clarity was an issue and asked for a variety of improvements in writing and presentation.  The authors seem to have addressed these comments in the discussion phase, therefore accept is "as is", but we would request the authors continue to carefully address the feedback for the camera ready version.

**Audience:**

Pruning networks is becoming more and more practically relevant as models are bigger models are being used more and more in real world applications.  Therefore understanding the effect that pruning has on model performance and e.g. their tendency to find spurious correlations is of great importance to ensure that these models remain robust.  The observation that pruning makes models more robust is perhaps surprising and certainly interesting broadly to the deep learning community.

**Claims And Evidence:**

The reviewers all found that the claims made in the submission were supported.  They noted that the empirical work was thorough and although it would've been exciting to see experiments with larger models, that seemed unreasonable given the cost of evaluating certified robustness.